# Advanced Glycation End Products Upregulate CD40 in Human Retinal Endothelial and Müller Cells: Relevance to Diabetic Retinopathy

**DOI:** 10.3390/cells13050429

**Published:** 2024-02-29

**Authors:** Jose-Andres C. Portillo, Amelia Pfaff, Sarah Vos, Matthew Weng, Ram H. Nagaraj, Carlos S. Subauste

**Affiliations:** 1Division of Infectious Diseases and HIV Medicine, Department of Medicine, Case Western Reserve University, Cleveland, OH 44106, USA; jcp31@case.edu (J.-A.C.P.); amelia.pfaff@case.edu (A.P.); sarah.vos@case.edu (S.V.); matthew.weng@case.edu (M.W.); 2Department of Ophthalmology, University of Colorado, Aurora, CO 80045, USA; ram.nagaraj@cuanschutz.edu; 3Department of Pathology, Case Western Reserve University, Cleveland, OH 44106, USA

**Keywords:** retina, diabetes, inflammation, adhesion molecule, chemokine, fibronectin, laminin

## Abstract

CD40 induces pro-inflammatory responses in endothelial and Müller cells and is required for the development of diabetic retinopathy (DR). CD40 is upregulated in these cells in patients with DR. CD40 upregulation is a central feature of CD40-driven inflammatory disorders. What drives CD40 upregulation in the diabetic retina remains unknown. We examined the role of advanced glycation end products (AGEs) in CD40 upregulation in endothelial cells and Müller cells. Human endothelial cells and Müller cells were incubated with unmodified or methylglyoxal (MGO)-modified fibronectin. CD40 expression was assessed by flow cytometry. The expression of ICAM-1 and CCL2 was examined by flow cytometry or ELISA after stimulation with CD154 (CD40 ligand). The expression of carboxymethyl lysine (CML), fibronectin, and laminin as well as CD40 in endothelial and Müller cells from patients with DR was examined by confocal microscopy. Fibronectin modified by MGO upregulated CD40 in endothelial and Müller cells. CD40 upregulation was functionally relevant. MGO-modified fibronectin enhanced CD154-driven upregulation of ICAM-1 and CCL2 in endothelial and Müller cells. Increased CD40 expression in endothelial and Müller cells from patients with DR was associated with increased CML expression in fibronectin and laminin. These findings identify AGEs as inducers of CD40 upregulation in endothelial and Müller cells and enhancers of CD40-dependent pro-inflammatory responses. CD40 upregulation in these cells is associated with higher CML expression in fibronectin and laminin in patients with DR. This study revealed that CD40 and AGEs, two important drivers of DR, are interconnected.

## 1. Introduction

Inflammation is central to the pathogenesis of complications of diabetes, including diabetic retinopathy and atherosclerosis. CD40, a member of the TNF receptor superfamily expressed on a broad range of cells, promotes these complications [1,2]. Studies in mouse models of diabetic retinopathy revealed that the expression of CD40 limited to non-hematopoietic cells is sufficient to promote this disorder [3,4]. Moreover, the engagement of CD40 by CD154 induces a broad range of pro-inflammatory responses in non-hematopoietic cells [5].

CD40 expression is elevated in endothelial cells and retinal Müller cells from mice and humans with diabetic retinopathy [2,6]. Moreover, a modest upregulation of CD40 causes a marked increase in CD40-driven pro-inflammatory responses in vitro [7]. Indeed, studies involving transgenic mice showed that after the induction of diabetes, the expression of CD40 restricted to endothelial or Müller cells was sufficient to induce inflammatory responses in the retina and, in the case of Müller cells, promote the development of diabetic retinopathy [3,4]. Taken together, these findings support that CD40 upregulation is an important event in the pathogenesis of diabetic retinopathy. However, it is not known what mediates the upregulation of CD40 in the diabetic retina.

Advanced glycation end products (AGEs) accumulate in tissues in diabetic patients and are linked to the development of complications of this disease [8,9]. AGEs are formed on amino groups (mainly those of lysine and arginine) of proteins via a multistep process of non-enzymatic glycosylation initiated by reducing sugars [9]. The reactive dicarbonyl intermediate methylglyoxal (MGO) is an important precursor of AGEs [8,10]. Plasma concentrations of MGO are elevated in diabetic patients [8,11,12]. Extracellular matrix proteins are particularly susceptible to AGE modification due to their long half-life [13].

AGEs accumulate in the retina at a high rate in diabetic patients [14,15]. AGE formation is prominent in capillaries of the diabetic retina, including the basal membrane and intracellularly in endothelial cells, pericytes, and smooth muscle cells [14,15,16]. AGEs also accumulate in Müller and ganglion cells in the retinas of diabetic rats [17]. Indeed, AGEs, especially *N^ε^*-carboxymethyl lysine (CML) adducts, have been detected not only in the retinal vasculature but also in neurosensory tissue of the diabetic eyes of rats and humans [18]. Moreover, AGEs appear to promote the development of diabetic retinopathy. Serum levels of AGEs, including CML, are increased in patients with diabetic retinopathy compared to diabetic patients without retinopathy [19]. Vitreal levels of AGEs are increased in patients with proliferative diabetic retinopathy compared to controls [19]. Incubation with AGEs increased vascular permeability in the retinal flat mounts of rats [20]. AGEs increase leukocyte adherence to retinal blood vessels and promote vascular leakage [21]. Inhibition in AGE formation through administration of aminoguanidine or pyridoxamine inhibits retinal pathologies in diabetic rodents [17,22]. In addition, attenuation of the effects of AGEs through administration of soluble receptors for AGEs reduces blood–retinal barrier breakdown, leukostasis, the expression of ICAM-1, neuronal dysfunction, and the development of capillary lesions in diabetic mice [23,24].

Herein, we report that AGE-modified fibronectin upregulates CD40 and potentiates the CD154-induced expression of ICAM-1 and CCL2 in retinal endothelial and Müller cells. Moreover, we report that areas of increased staining with an anti-CML antibody in the extracellular matrix proteins, fibronectin and laminin, are associated with increased expression of CD40 in endothelial and Müller cells in the retinas of patients with diabetic retinopathy. These results suggest that AGEs are important drivers of CD40 upregulation in endothelial and Müller cells and promote CD40-mediated inflammation.

## 2. Materials and Methods

### 2.1. Cells

Two different batches of primary human aortic endothelial cells were obtained from Lonza (Walkersville, MD, USA) and were cultured following the manufacturer’s recommendations. Primary human retinal endothelial and Müller cells were obtained as described [2]. Retinal endothelial cells were cultured in DMEM plus 10% FBS (HyClone, Logan, UT, USA), endothelial cell growth supplement from bovine pituitary extract (15 g/mL; Sigma Chemical, St. Louis, MO, USA), and insulin/transferrin/selenium (Sigma-Aldrich, St Louis, MO, USA). Cell identity was confirmed by incorporation of acetylated low-density lipoprotein (>90%) and staining with an antibody for von Willebrand factor. Primary human Müller cells were cultured in DMEM/F12 containing 20% FBS. Cultures were >95% pure for Müller cells (vimentin^+^, CRALBP^+^, and GFAP^–^ by immunofluorescence). Cells were used between passages 3 to 6. These cells were isolated from 3 to 4 different donors.

### 2.2. In Vitro Stimulation

Tissue culture plates were coated with fibronectin (2 μg/cm^2^) and incubated at room temperature for 1 h. This was followed by incubation with sterile solutions of methylglyoxal (MGO; Sigma-Aldrich, St. Louis, MO, USA) in PBS or PBS alone at 37 °C for 1 week. Human serum albumin (Sigma-Aldrich) was also incubated with or without MGO. MGO was purified by two cycles of distillation under low pressure. Pierce^TM^ columns (Thermo-Fisher, Waltham, MA, USA) were used to remove potential endotoxin contamination in MGO. In certain experiments, fibronectin was incubated with MGO in the presence of aminoguanidine (500 μM; Sigma-Aldrich). The plates were washed three times with PBS to remove any unbound MGO before adding cells. Cells were incubated with control or MGO-modified proteins for 48 h. This was followed by stimulation for 24 h with multimeric human CD154 (CD40 ligand; gift from Dr. Richard Kornbluth, Multimeric Biotherapeutics Inc., La Jolla, CA, USA) [2]. We used incubation with a non-functional CD154 mutant (T147N) as the control. In certain experiments, cells were incubated with IFN-γ (100 U/mL; Peprotech, Rocky Hill, NJ, USA) plus TNF-α (100 ng/mL; Peprotech) for 24 h.

### 2.3. Flow Cytometry

Cells were stained with anti-CD40, anti-ICAM-1 (eBiosciences, San Diego, CA, USA), or isotype control mAbs. Cells were analyzed on an LSR II flow cytometer (Becton Dickinson, San Jose, CA, USA). FlowJo software version 10.10 (Tree Star Inc., Ashland, OR, USA) was used for analysis.

### 2.4. ELISA

ELISA for hydroimidazolone (HI) was performed as described [25]. Microplate wells were coated with fibronectin or MGO-modified fibronectin followed by incubation with anti-HI monoclonal antibody. Wells were incubated with a secondary antibody followed by the addition of tetramethylbenzidine substrate (Sigma-Aldrich). The reaction was stopped by using 2N H_2_SO_4_, and the absorbance was measured at 450 nm in a Dynex MRX 5000 Microplate Reader. Human CCL2 was measured using an ELISA kit (R&D Systems, Minneapolis, MN, USA).

### 2.5. Human Subjects

Posterior poles were obtained postmortem from eight donors with diabetic retinopathy and three non-diabetic control individuals (Eversight, Cleveland, OH, USA). There was no history of any other retinal diseases in these donors. Four diabetic subjects had proliferative diabetic retinopathy. No information regarding the disease stage was available from the other donors. Posterior poles were fixed in 4% paraformaldehyde within 16 h of death and were kept in paraformaldehyde for more than 24 h. The use of human material was in accordance with the Declaration of Helsinki on the use of human material for research.

### 2.6. Immunohistochemistry

Frozen sections were incubated with anti-*N^ε^*-carboxymethyl lysine (CML) monoclonal antibody (CML26, Novus Biologicals, Littleton, CO, USA), anti-fibronectin antibody (Bioss, Woburn, MA, USA), anti-laminin antibody (Bioss), anti-human CD40 antibody (Bioss), anti-vimentin antibody (Novus Biologicals), or anti-von Willebrand factor antibody (GeneTex, Irvine, CA, USA). Fluorescent secondary antibodies were from Jackson ImmunoResearch Laboratories, West Grove, PA. Antibodies were used at the manufacturer-recommended dilutions. Staining specificity was confirmed by omitting the primary antibody. Retinas were analyzed blindly in an Olympus FV1200 IX-83 confocal microscope. Images were processed in Photoshop CC 19.1.1. using similar linear adjustments for all samples. Semiquantitative assessment of CML and CD40 expression was performed using MetaMorph (Nashville, TN, USA) [6]. Briefly, areas positive for von Willebrand factor and fibronectin or vimentin and laminin were identified. Pixel intensity for CML in the selected areas was measured. CD40 expression was measured in areas positive for Willebrand factor and CML or vimentin and CML.

### 2.7. Statistical Analysis

All results were expressed as the mean ± SD or SEM. Data were analyzed using one-way ANOVA followed by Bonferroni correction. Differences were considered statistically significant at *p* ≤ 0.05.

## 3. Results

### 3.1. Methylglyoxal-Modified Fibronectin and Albumin Upregulate CD40 Expression in Human Endothelial Cells

Fibronectin was used to examine the effects of AGE-induced protein modification because fibronectin is an important component of the basal membrane of retinal capillaries that accumulates AGEs [26]. Tissue culture plates coated with fibronectin were incubated with PBS or increasing concentrations of MGO, a major precursor of AGEs. MGO caused a significantly increased detection of hydroimidazolone (HI), a prominent protein modification induced by MGO [27] (Figure 1A). Moreover, HI detection was inhibited by incubation with aminoguanidine, a hydrazine compound that prevents AGE formation [28] (Figure 1A). We plated primary human aortic endothelial cells onto wells coated with fibronectin pre-treated with PBS or MGO (100 μM). Cells were also cultured with IFN-γ and TNF-α as a positive control. Culturing cells on MGO-modified fibronectin caused a significant upregulation of CD40 that was ablated by aminoguanidine (Figure 1B). Fibronectin incubated with MGO (100 μM) upregulated CD40 in human retinal endothelial cells (Figure 1C). We also examined the effect of human serum albumin incubated with MGO (100 μM) since albumin is one of the main proteins subjected to AGE modification due to its abundance [29]. Human serum albumin incubated with MGO also upregulated CD40 in human retinal endothelial cells (Figure 1D). Thus, MGO-derived AGEs upregulate CD40 in primary endothelial cells.

### 3.2. MGO-Modified Fibronectin Upregulates CD40 in Primary Human Retinal Müller Cells

CD40 expressed in Müller cells plays a central role in the induction of inflammatory responses and the development of diabetic retinopathy [3]. Moreover, the expression of CD40 in Müller cells is increased in mice and patients with diabetic retinopathy [2,6]. We examined the effect of MGO-modified fibronectin on the expression of CD40 in Müller cells. Incubation of primary human Müller cells with MGO-modified fibronectin upregulated the expression of CD40 (Figure 2).

### 3.3. MGO-Modified Fibronectin Enhances Pro-Inflammatory Responses Triggered by CD40 Ligation

We examined the functional relevance of CD40 upregulation induced by MGO-modified fibronectin. ICAM-1 upregulation is an important pro-inflammatory response induced by CD154 in endothelial cells. First, we identified a dilution of the CD154 supernatant that would induce submaximal pro-inflammatory molecule upregulation. Incubation of retinal endothelial cells with a 1:90 dilution of CD154 supernatant was sufficient to induce a significant but submaximal upregulation of ICAM-1 (Figure 3A). Human retinal endothelial cells incubated with fibronectin or MGO-modified fibronectin were stimulated with or without CD154. CD154-driven upregulation of ICAM-1 was significantly higher in endothelial cells incubated with MGO-modified fibronectin compared to that of cells incubated with naive fibronectin (Figure 3B). Similarly, CCL2 secretion was significantly higher in endothelial cells (Figure 3C) and Müller cells (Figure 3D) incubated with MGO-modified fibronectin compared to naive fibronectin. Taken together, MGO-modified fibronectin not only upregulates CD40 but also potentiates CD40-driven pro-inflammatory responses in retinal endothelial and Müller cells.

### 3.4. Increased Staining with an Anti-CML Antibody in Extracellular Matrix Proteins Is Associated with Areas of Increased CD40 Expression in Endothelial and Müller Cells in the Retinas of Patients with Diabetic Retinopathy

The expression of CD40 is elevated in endothelial and Müller cells in the retinas from patients with diabetic retinopathy [6]. Moreover, areas of increased CD40 expression co-localize with pro-inflammatory molecules (ICAM-1 in endothelial cells and CCL2 in Müller cells) [6]. To assess the in vivo relevance of AGEs in CD40 upregulation, we examined AGE-induced protein modification and CD40 expression in posterior poles from patients with diabetic retinopathy. These included four subjects with a history of proliferative diabetic retinopathy and four subjects with a prior diagnosis of diabetic retinopathy but without available information on the stage of disease (Table 1). These subjects were labeled as having diabetic retinopathy. We examined three non-diabetic subjects without a history of retinal disease as controls.

To assess AGE-induced protein modification, we utilized an antibody against CML (Appendix A, Appendix A), an AGE with a high prevalence of accumulation in proteins in diabetes [30,31]. Antibodies against von Willebrand factor and vimentin were used to identify endothelial and Müller cells, respectively. Fibronectin was co-expressed with von Willebrand factor consistent with the previous finding that endothelial cells are coated with fibronectin [32] (Figure 4A). Fibronectin was also detected surrounding retinal endothelial cells (Figure 4A). Compared to non-diabetic subjects, patients with diabetic retinopathy exhibited significantly increased staining with the anti-CML antibody that was associated with fibronectin and endothelial cells (Figure 4A). Laminin is another critical extracellular matrix protein that accumulates AGEs in diabetes [33]. Consistent with a previous study [34], laminin was detected in areas that included those surrounding blood vessels and where Müller cell end feet are located in the inner retina (Figure 4B). Compared to non-diabetic subjects, patients with diabetic retinopathy had significantly more intense staining with the anti-CML antibody in areas that co-expressed laminin associated with Müller cells in the inner retina and laminin surrounding blood vessels (Figure 4B).

We examined CD40 expression in endothelial and Müller cells. Endothelial cells from patients with diabetic retinopathy exhibited increased CD40 expression that was associated with increased CML staining in these cells or the presence of CML surrounding endothelial cells (Figure 5A). Similarly, Müller cells from these patients exhibited increased staining for CD40 that was associated with CML staining in these cells or areas of CML staining in the retina (Figure 5B). Together, increased staining with the anti-CML antibody in extracellular matrix proteins in patients with diabetic retinopathy was associated with increased CD40 expression in endothelial and Müller cells.

## 4. Discussion

Increased expression of CD40 is a hallmark of CD40-driven diseases [35,36]. CD40 upregulation is relevant because it promotes pro-inflammatory responses downstream of this receptor [7]. We report that proteins containing AGE-induced modifications increase the expression of CD40 in human endothelial cells and Müller cells. AGE-mediated CD40 upregulation was functionally relevant since it enhanced CD40-driven expression of ICAM-1 and production of CCL2, events central to the pathogenesis of inflammatory disorders, including diabetic retinopathy. Immunohistochemistry studies revealed areas of increased staining with the anti-CML antibody in extracellular matrix proteins in the retinas of patients with diabetic retinopathy, and these areas were associated with increased CD40 expression in retinal endothelial and Müller cells. Altogether, these results support that AGEs promote CD40 upregulation in the setting of diabetes and potentiate the inflammatory effects of CD40.

Staining with the anti-CML antibody in fibronectin and laminin around retinal capillaries is suggestive of AGE-induced protein modifications that are reported to occur in the basal membranes in the retina [15]. AGE-mediated crosslinking of extracellular matrix proteins increases vascular stiffness and induces apoptosis of pericytes, resulting in vascular dysfunction that promotes diabetic retinopathy [37,38,39]. Our studies suggest that AGE-induced modification of extracellular matrix proteins increased CD40 expression in the diabetic retina since increased expression of CD40 in endothelial cells is associated with CML-expressing fibronectin, and fibronectin containing AGE-induced modification upregulates CD40 in endothelial cells. While fibronectin is an important target for AGE modification in the diabetic retina [26], other proteins present in the extracellular matrix such as laminin and collagen can accumulate AGEs [40]. Indeed, we detected increased staining with the anti-CML antibody in laminin surrounding retinal blood vessels, and it is associated with Müller cells. Moreover, relevant to the effects of AGEs on CD40 expression in Müller cells, AGE-modified laminin downregulates Kir4.1 expression in these cells due to the disorganization of the actin cytoskeleton and disruption of α-dystroglycan-syntrophindystrophin complexes [41]. Our studies centered on extracellular proteins since they are major targets of AGE modification. However, AGE-induced modifications can also accumulate intracellularly in cells exposed to high concentrations of glucose [42].

AGEs have been linked to various inflammatory responses that are key to the development of diabetic retinopathy. These include ICAM-1 upregulation, leukostasis, vascular leakage, apoptosis, and the development of capillary lesions [20,21,23,24,39]. Interestingly, CD40 promotes ICAM-1 upregulation, leukostasis, and retinal capillary degeneration, and is required for the development of diabetic retinopathy [2]. Our studies suggest that AGEs promote CD40 upregulation, which raises the possibility that CD40 and AGEs work together to induce events critical for the development of diabetic retinopathy.

The current work centered on Müller and endothelial cells because CD40 expressed in non-hematopoietic cells is central to the development of inflammation, and the expression of CD40 restricted to Müller or endothelial cells is sufficient to induce retinal inflammation in the setting of diabetes [3,4]. Given that AGEs can upregulate CD40 in monocytes [43], it is possible that AGEs can also promote inflammatory responses induced by CD40 ligation in retinal microglia/macrophages.

In addition to diabetic retinopathy, previous immunohistochemical studies have revealed the presence of AGE-induced modifications in other tissues affected by diabetes. These include the kidney, peripheral nerves, and atherosclerotic lesions of arterial walls [44,45,46]. Indeed, AGEs are also considered to be involved in the pathogenesis of complications in diabetes at these sites [47,48,49]. CD40 is also associated with the development of diabetic kidney disease, atherosclerosis, and potentially, peripheral neuropathy [1,50,51]. Thus, the link between AGEs and CD40 may be relevant to various complications of diabetes.

In summary, we report that CD40 and AGEs, two important drivers of diabetic retinopathy, are interrelated at the level of retinal cells in the diabetic retina. Despite the various factors that promote the development of diabetic retinopathy, genetic disruption of CD40 is sufficient to prevent the development of this disease, and pharmacologic inhibition of CD40 signaling ablates inflammation in the diabetic retina [2,4]. Our findings suggest an interconnection between CD40 and AGEs that may help explain why targeting a single pathogenic pathway effectively controls diabetic retinopathy.

## Figures and Tables

**Figure 1 cells-13-00429-f001:**
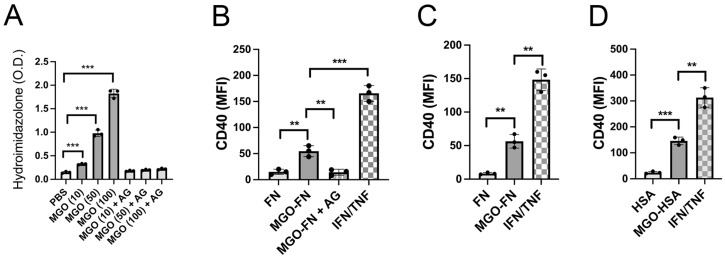
Methylglyoxal-modified fibronectin or albumin upregulate CD40 in human endothelial cells. (**A**) Fibronectin was incubated for 7 days with increasing concentrations of MGO (10–100 μM) with or without aminoguanidine. Hydroimidazolone was detected by ELISA. Each group consisted of 3 different wells. (**B**,**C**) Primary human aortic endothelial cells (**B**) or human retinal endothelial cells (**C**) were cultured for 48 h in wells coated with fibronectin (FN) or fibronectin pretreated with either MGO or MGO plus aminoguanidine (AG). IFN-γ (IFN, 100 U/mL) and TNF-α (100 ng/mL) were used as positive controls. CD40 was assessed by flow cytometry. (**D**) Retinal endothelial cells were incubated with human serum albumin (HSA) or MGO-modified albumin followed by assessment of CD40 expression. Each group consisted of 3 individual replicates. Results are shown as mean ± SD and are representative of 3 independent experiments using cells from 2 to 3 different donors. ** *p* < 0.01, *** *p* < 0.001.

**Figure 2 cells-13-00429-f002:**
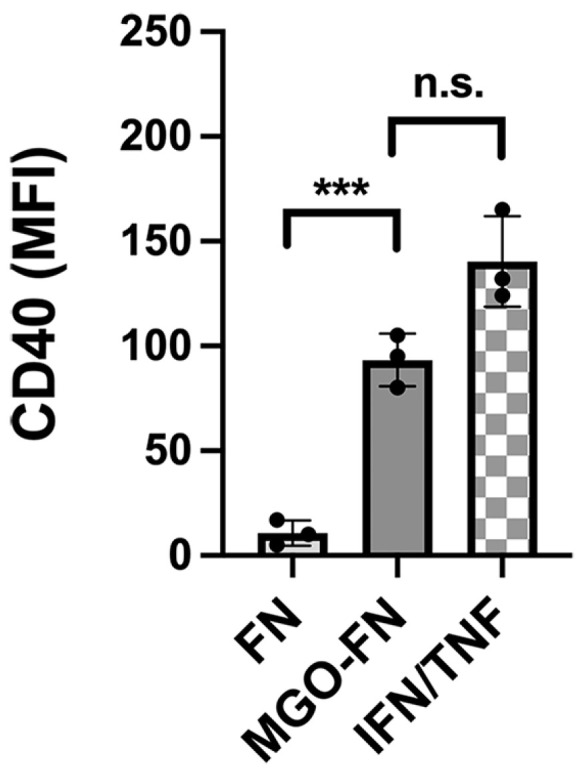
Methylglyoxal-modified fibronectin upregulates CD40 expression in Müller cells. Primary human Müller cells were cultured in wells coated with fibronectin or MGO-modified fibronectin as above. IFN-γ and TNF-α were used as positive controls. CD40 was assessed by flow cytometry. Each group consisted of 3 individual replicates. Results are shown as mean ± SD and are representative of 3 independent experiments using cells from 3 different donors. *** *p* < 0.001. n.s. indicates non-significant statistical difference.

**Figure 3 cells-13-00429-f003:**
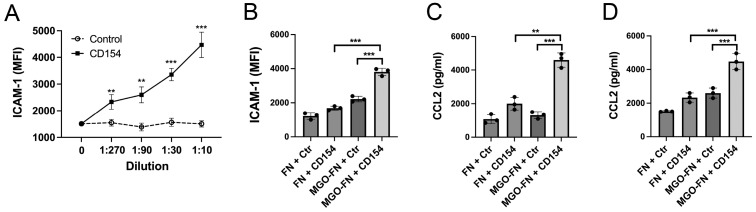
Methylglyoxal-modified fibronectin potentiates CD154-induced upregulation of ICAM-1 and CCL2 in human retinal endothelial cells and Müller cells. (**A**) Primary human retinal endothelial cells were stimulated for 24 h with varying dilutions of CD154. ICAM-1 was examined by flow cytometry. (**B**) Primary human retinal endothelial cells incubated with fibronectin (FN) or MGO-modified fibronectin for 48 h were stimulated with a suboptimal dilution of CD154 (1:90). ICAM-1 was assessed by flow cytometry after 24 h. (**C**) Primary human endothelial cells incubated with fibronectin or MGO-modified fibronectin were stimulated with a suboptimal dilution of CD154 (1:90). CCL2 production was assessed by ELISA after 24 h. (**D**) Primary human Müller cells incubated with fibronectin or MGO-modified fibronectin were stimulated with CD154 (1:90). CCL2 production was assessed by ELISA after 24 h. Each group consisted of 3 individual replicates. Results are shown as mean ± SD and are representative of 3 independent experiments using cells from 3 different donors. ** *p* < 0.01; *** *p* < 0.001.

**Figure 4 cells-13-00429-f004:**
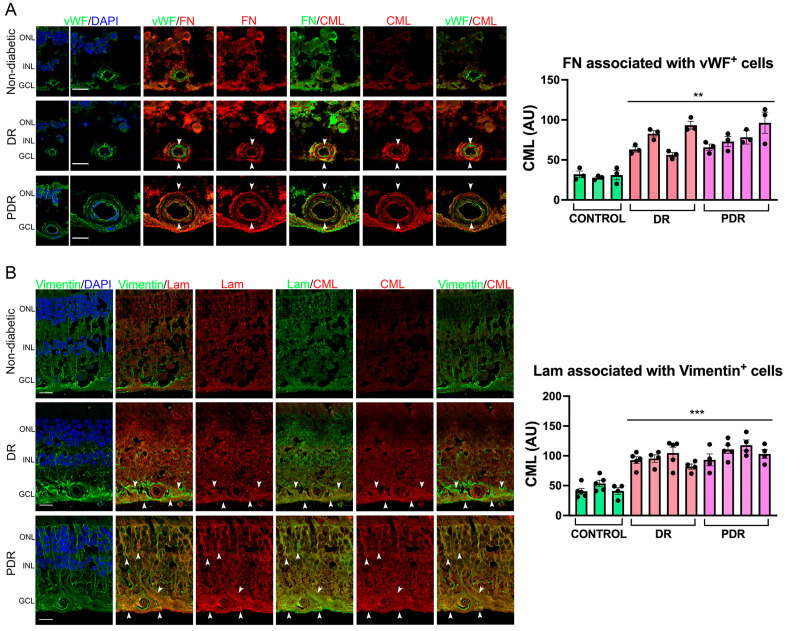
Patients with diabetic retinopathy exhibit increased expression of CML in fibronectin and laminin that associates with retinal endothelial and Müller cells. (**A**) Posterior poles from patients with diabetic retinopathy (DR; no information about disease stage was available), proliferative diabetic retinopathy (PDR), and non-diabetic controls were incubated with antibodies against von Willebrand factor (vWF; endothelial cell marker), CML, and fibronectin (FN), followed by secondary antibodies conjugated with Alexa Fluor 488, Alexa Fluor 568, and Alexa Fluor 647, respectively. Fibronectin was pseudo-colored red or green as indicated. Arrowheads show areas of fibronectin that co-express von Willebrand factor and fibronectin which surround retinal endothelial cells. These areas also display intense CML expression. Original magnification ×400. Scale bar, 20 μm. (**B**) Posterior poles were incubated with antibodies against vimentin (Müller cells), CML, and laminin (Lam) followed by secondary antibodies conjugated with Alexa Fluor 488, Alexa Fluor 568, and Alexa Fluor 647, respectively. Laminin was pseudo-colored red or green as indicated. Laminin is detected in the inner retina in areas that co-express vimentin (arrowheads) as well as surrounding structures that appear to be blood vessels. Intense CML staining is co-expressed with laminin in subjects with diabetic retinopathy. Areas of CML staining that co-express laminin and vimentin are also detected in the outer nuclear layer of the subject with PDR (arrowheads). Scale bar, 25 μm. GCL, ganglion cell layer; INL, inner nuclear layer; ONL, outer nuclear layer. Graphs show pixel intensity for CML (arbitrary units [AUs]) in fibronectin associated with von Willebrand factor, and laminin associated with vimentin. Three to five fields per subject were analyzed. The disease stage was not known in 4 diabetic subjects (DR). No comparison between the 2 groups of diabetic patients was performed because subjects labeled as having diabetic retinopathy may have had proliferative diabetic retinopathy. Comparison was performed between subjects without diabetic retinopathy and the 8 subjects diagnosed with some form of diabetic retinopathy. Data shown are mean ± SEM. Statistical comparison was conducted using the mean values. ** *p* < 0.01; *** *p* < 0.001.

**Figure 5 cells-13-00429-f005:**
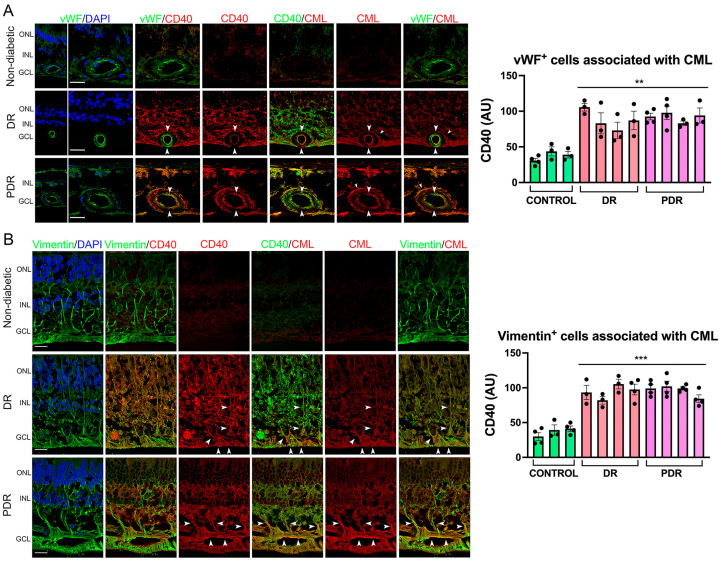
Patients with diabetic retinopathy exhibit increased expression of CD40 in endothelial and Müller cells that associates with areas of increased expression of CML. (**A**) Posterior poles from patients with diabetic retinopathy (DR), proliferative diabetic retinopathy (PDR), and non-diabetic controls were incubated with antibodies against von Willebrand factor (vWF), CML, and CD40 followed by secondary antibodies conjugated with Alexa Fluor 488, Alexa Fluor 568, and Alexa Fluor 647, respectively. CD40 was pseudo-colored red or green as indicated. Arrowheads show endothelial cells with enhanced CD40 staining that co-express CML. Arrows show areas of intense CML expression that surround CD40-expressing endothelial cells. Original magnification ×400. Scale bar, 20 μm. (**B**) Posterior poles were incubated with antibodies against vimentin, CML, and CD40 followed by secondary antibodies conjugated with Alexa Fluor 488, Alexa Fluor 568, and Alexa Fluor 647, respectively. CD40 was pseudo-colored red or green as indicated. Arrowheads show Müller cells where increased CD40 expression associates with CML. Scale bar, 25 μm. GCL, ganglion cell layer; INL, inner nuclear layer; ONL, outer nuclear layer. Graphs show pixel intensity for CD40 (arbitrary units [AUs]) in endothelial cells associated with CML, and Müller cells associated with laminin. Three to five fields per subject were analyzed. Subjects without diabetic retinopathy were compared to the 8 subjects diagnosed with some form of diabetic retinopathy. Statistical comparison among patient samples was conducted using the mean values for these measurements. Data shown are mean ± SEM. ** *p* < 0.01; *** *p* < 0.001.

**Table 1 cells-13-00429-t001:** Human subjects.

Diagnosis	Age/Sex	Other Clinical Information
Diabetic retinopathy	58/M	Gangrene and stroke
Diabetic retinopathy	71/M	Myocardial infarction, end-stage renal disease, diabetic neuropathy, gastro-intestinal bleed, and hyperlipidemia
Diabetic retinopathy	73/F	Myocardial infarction, heart failure, hypertension, and hyperlipidemia
Diabetic retinopathy	69/M	Myocardial infarction, ischemic cardiomyopathy, hypertension, stroke, and hyperlipidemia
Proliferative diabetic retinopathy	73/M	Gastrointestinal bleed, myocardial infarction, and hyperlipidemia
Proliferative diabetic retinopathy	73/M	Myocardial infarction, chronic kidney disease, hypertension, and hyperlipidemia
Proliferative diabetic retinopathy	63/M	Coronary artery disease, heart failure, hypertension, and pneumonia
Proliferative diabetic retinopathy	58/F	Myocardial infarction, heart failure, chronic kidney disease, and hyperlipidemia
Non-diabetic	78/M	Myocardial infarction, hypertension, hyperlipidemia, and alcoholism
Non-diabetic	61/M	Myocardial infarction, hyperlipidemia, and hypertension
Non-diabetic	53/M	Lung cancer and hypertension

## Data Availability

All research data are presented in the manuscript.

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
