# Peer review of "Advanced Glycation End Products Upregulate CD40 in Human Retinal Endothelial and Müller Cells: Relevance to Diabetic Retinopathy"

_cells, 2024, doi:10.3390/cells13050429_

Round 1

Reviewer 1 Report

Comments and Suggestions for Authors

The article titled “ Advanced Glycation end Products Upregulate CD40 in Human Retinal Endothelial and Müller Cells: Relevance to Diabetic Retinopathy  by Jose-Andres C. Portillo  et al.

analyzes the role of advanced glycation end products (AGEs) in CD40 upregulation in endothelial cells and Müller cells in vitro approaches, and the AGEs presence in endothelial and Müller cells from patients. The results showed that AGEs is an inducers of CD40 upregulation in endothelial and Müller cells promoted the pro-inflammatory responses.

Some comments:

1.In the description of the results in Figures 1 and 2, please indicate that the INFgamma+TNFalpha cocktel is used as a positive control in the text and not in the footnote and include the significant differences obtained compared to the MGO-FN treatment. Similarly, comment on the HSA treatment in the main text of results.

2. Page 5 line 197, put figure 3C behind endothelial, and figure 3D behind Müller cells.

3. It would be interesting to know the expression of CD40 in induction with CD154 upon the experimental condition showed.

4. Figures 4 and 5 do not describe the results obtained in immunolabeling comparatively between RD and RDP patients

5. The HSA data shown is reduced to a graph, I do not consider that the abstract should indicate that the work has been done analyzing the effects of HSA.

In general the results shown are briefly described, and with no comparison with the conditions shown. Similarly in the discussion.

Author Response

Reviewer 1 

  1. As recommended by the reviewer, we now state in the Results section that IFN-g plus TNF-a were used as positive controls (see page 4-5, lines 162-163). In addition, we also include statistical analysis comparing the effects of MGO with those of IFN-g plus TNF-a (Figures 1 and 2). Following the reviewer’s recommendation, we now explain why we included work with human serum albumin (page 5, lines 166-168). We also provide rationale for the work with fibronectin (page 4, lines 154-156).

  1. As recommended, we placed endothelial cells prior to Figure 3C and Müller cells prior to Figure 3D (page 6, lines 204-205).

  1. The reviewer asked about the expression of CD40 in the experiments with CD154 stimulation. We did not examine CD40 expression since the presence of CD154 may interfere with the binding of anti-CD154 monoclonal antibodies to CD154.

  1. The reviewer commented that we did not compare findings in patients labelled as having diabetic retinopathy with those in patients with prior diagnosis of proliferative diabetic retinopathy (Figures 4 and 5). We have included semiquantitative analysis of the staining with anti-CML and anti-CD40 antibodies (Figures 4 and 5). We did not make any formal comparisons between the 2 groups because it is possible that at least some of the patients in the diabetic retinopathy group had proliferative diabetic retinopathy. The diabetic retinopathy group consisted of patients with no clinical information on the stage of the disease. This is now explained in the legends for Figures 4 and 5.

  1. As requested by the reviewer, we no longer refer to studies with HSA in the abstract.

  1. The reviewer commented that there was no comparison between the groups of patients in the Results and Discussion sections. Please see #4.

Reviewer 2 Report

Comments and Suggestions for Authors

The authors performed studies to examine the role of advanced glycation end products (AGEs) in upregulation of CD40 in in endothelial cells and Müller cells during diabetic retinopathy. Human endothelial cells and Müller cells were incubated with unmodified or methylglyoxal (MGO)-modified fibronectin or albumin. CD40 expression was assessed by flow cytometry. Expression of ICAM-1 and CCL2 were examined by flow cytometry or ELISA after stimulation with CD154 (CD40 ligand). Expression of carboxymethyl lysine (CML), fibronectin, and laminin as well as CD40 in endothelial and Müller cells from patients with DR were examined by confocal microscopy. Their data show that fibronectin and albumin modified by MGO upregulated CD40 in endothelial and Müller cells. MGO-modified fibronectin enhanced CD154-driven upregulation of ICAM-1 and CCL2 in endothelial and Müller cells. Increased CD40 expression in endothelial and Müller cells from patients with DR was associated with increased carboxymethyl lysine ( CML) co-localization with fibronectin and laminin. These studies identified AGEs as inducers of CD40 upregulation in endothelial and Müller cells and enhancers of CD40-dependent pro-inflammatory responses. CD40 upregulation in these cells associated with higher CML co-localization with fibronectin and laminin in patients with DR. Based on these observations the authors conclude that CD40 and AGEs are interconnected during DR.

These results are potentially interesting. However, there are a few problems with the manuscript as presented.

The use of 2-tailed Student’s t-test to analyze differences between more than 2 groups is not acceptable. Group differences between three or more groups should be tested using ANOVA and appropriate post-hoc comparisons.

The data are reported as representative of 3 independent experiments, but the number of samples compared is not provided. Tissue culture studies should be repeated with 3 different batches of cells and the n value in each independent experiment should be provided in the figure legends. In other words, how many samples were compared in the representative independent experiment shown in the figure?

Please provide a table listing antibodies with their suppliers and catalogue numbers.

The terminology used to describe fibronectin and laminin modification by methyglyoxal treatment is misleading. The AGE-induced increases in HI levels by ELISA and CML levels by immunolocalization reflect AGE-induced increases in protein modification rather than increases in HI or CML expression as stated repeatedly in the text (lines 81, 152, 213, 228, 234, 238,242, 250, 263, 269, 306)

The following sentence (lines 328-329) may be misleading and should be rephased: “In addition to diabetic retinopathy, immunohistochemical studies revealed the presence of AGEs in other tissues affected by diabetes.”  Presumably, the authors mean that previous immunohistochemical studies have revealed….etc.

Author Response

Reviewer 2:

  1. As recommended by the reviewer, that data were analyzed using ANOVA followed by Bonferroni correction (see page 4, lines 148-149).

  1. The data are reported as representative of 3 independent experiments, but the number of samples compared is not provided. Tissue culture studies should be done with 3 different batches of cells and the n value in each independent experiment should be provided in the figure legends. In other words, how many samples were compared in the representative independent experiment shown in the figure?

      We clarified in the legends for Figures 1 – 3 that the data shown originate              from 3 batches of cells. These cells were grown and stimulated separately.

  1. As requested, we now provide a table listing antibodies with their suppliers and catalogue numbers (see Appendix; Table 2).

  1. As recommended by the reviewer, we removed statements about levels of HI or CML. We now refer to increases in AGE-induced protein modification or in staining with anti-CML antibody. We hope that the reviewer finds those changes acceptable.

  1. Following the reviewer’s advice, we now state that we refer to previous immunohistochemical studies (page 11, line 345).

Reviewer 3 Report

Comments and Suggestions for Authors

The manuscript entitled “Advanced Glycation end Products Upregulate CD40 in Human 2 Retinal Endothelial and Müller Cells: Relevance to Diabetic 3 Retinopathy” identified AGEs as inducers of CD40 upregulation in endothelial and Müller cells and enhancers of CD40-dependent pro-inflammatory responses. CD40 upregulation in these cells is associated with higher CML expression in fibronectin and laminin in patients with Diabetic Retinopathy. These studies revealed that CD40 and AGEs, two important drivers of Diabetic Retinopathy, are interconnected, summarizes the functions of ILCs in the context of the immunology of infections caused by different intracellular and extracellular pathogens and discusses their possible therapeutic potential”. The manuscript needs major revision before it can be accepted for publication.

1.     The concentration of Methylglyoxal used is the study is 0, 10 and 100 mM. Are these concentrations fall in physiological range? If not, why the authors have not used physiological concentration of MG as that will make the study more appropriate.

2.     Methylglyoxal is a strong glycating agent. Are there other glycating agents besides MG that reportedly cause CD40 upregulation?

3.     In patients study there are only 4 patients of diabetic retinopathy and 4 of proliferative DR based on which the authors have concluded that increased expression of CML in extracellular matrix proteins is associated with areas of increased CD40 expression in endothelial and muller cells in the retina of patients with Diabetic Retinopathy. Four patients is a low number to come to any conclusion. The authors should have at least 10 patients of each type to reach to any conclusion.

4.     The authors need to add a few more recent references relevant to the work.

Comments on the Quality of English Language

There is minor edits needed to make english more appropriate

Author Response

Reviewer 3:

  1. The reviewer asked whether the concentrations of Methylglyoxal used in the study were within a physiological range.

This is indeed the case. Concentrations within the micromolar range approximate reported physiological levels. MGO concentrations as high as 300 mM have been reported in mammalian cells (Chaplen et al. PNAS 1998; 95:5533).  Moreover, it is likely that actual levels may be higher than reported since MGO is a highly reactive compound (Liu et al. IOVS 2004; 45:1983).

  1. Methylglyoxal is a strong glycating agent. Are there other glycating agents besides MGO that reportedly cause CD40 upregulation?

Takahashi et al (reference 43) reported that glyceraldehyde-derived AGE and glycolaldehyde-derived AGE also upregulate CD40 in peripheral blood human monocytes.

  1. There are only 4 patients of diabetic retinopathy and 4 of proliferative DR based on which the authors have concluded that increased expression of CML in extracellular matrix proteins is associated with areas of increased CD40 expression in endothelial and muller cells in the retina of patients with Diabetic Retinopathy. Four patients is a low number to come to any conclusion.

There was clinical information on the stage of diabetic retinopathy in 4 subjects. These patients carried a prior diagnosis of proliferative diabetic retinopathy. The disease stage was not known in the remaining 4 diabetic subjects. We did not compare the 2 groups of diabetic patients because some of the subjects labelled as having diabetic retinopathy may have had proliferative diabetic retinopathy. Instead, we compared non-diabetic subjects to the 8 subjects that had been diagnosed with some form of diabetic retinopathy. We have included semiquantitative assessment of staining with anti-CML or anti-CD40 mAbs (Figures 4 and 5). These analysis supports that, compared to the non-diabetic subjects, the 8 subjects with diabetic retinopathy exhibited increased staining with anti-CML Ab in fibronectin and laminin that were associated with increased CD40 expression in retinal endothelial and Muller cells. Finally, we should point out that it is difficult to obtain posterior poles from subjects with diabetic retinopathy. It took us two and a half years to obtain the posterior poles used in this study.

  1. We added a few more recent references relevant to the work (see ref. 5, 8, 9, 13).

Round 2

Reviewer 1 Report

Comments and Suggestions for Authors

The article has been improved with the different reviewer´s suggestions.

Author Response

We are pleased to see that the reviewer considers that the article has been improved and has no further comments.

Reviewer 2 Report

Comments and Suggestions for Authors

The authors have addressed most of my previous concerns. However, they appear to have misunderstood a major concern about the need for replication of tissue culture experiments with different batches of cells. 

As was stated in my previous review, “Tissue culture studies should be repeated with 3 different batches of cells and the n value in each independent experiment should be provided in the figure legends.”

However, they added the following statement in the figure legends: “Each group consisted of 3 different batches of cells”. Presumably this means they compared 3 different wells plated from the same batch of cells in a single experiment. This is not acceptable. In tissue culture experiments, the results should always be verified with at least one additional independent batch of cells.

The statistical comparisons in Figures 4 and 5 are unclear. An indication of the non-significant (ns) differences within the control and diabetic groups should be added if the differences within those groups are not significant.

Author Response

  1. The reviewer raised concern about the need for replication of tissue culture experiments with different batches of cells.

The retinal endothelial and Müller cells used in this study were isolated by Dawn Smith, manager of the Analytical Services Core at the Visual Science Core Center at Case Western Reserve University. She isolated endothelial and Müller cells from 3 to 4 different donors. These different batches of primary cells were used for our studies. We used 2 batches of commercial human aortic endothelial cells for this work: one that had been obtained for our previous published work with this type of cells and a second batch that was purchased when this project was ongoing. This information is now stated in the Materials and Methods section (page 3, lines 85 and 95). As recommended by the reviewer, we now clarify in the legends for Figures 1 to 3 that studies were conducted using cells from different donors.

  1. The reviewer requested adding an indication of non-significant (ns) differences within the control and diabetic groups.

We stated in the Results section (page 7 lines 244-245 and 249-250) and in the legends for Figures 4 and 5 (lines 274-275, 300-301) that we were comparing non-diabetic subjects to those with diabetic retinopathy. We now include brackets below the x-axis to help identify the groups of subjects. We now show only 1 horizontal line and this is located above the bars for the subjects with diabetic retinopathy. This line is accompanied by the appropriate number of asterixis. We hope that this helps clarify that we were comparing the whole group of diabetics to the group of control subjects. We did not include statical analysis within the group of controls and within the group of diabetics since that was not the purpose of the experiments. We hope that the reviewer finds the modifications to the figures acceptable.

Reviewer 3 Report

Comments and Suggestions for Authors

The comments have been addressed well.

Author Response

We are pleased that the reviewer considers that the comments have been addressed appropriately

Round 3

Reviewer 2 Report

Comments and Suggestions for Authors

The revisions have addressed my concerns.